# Incidents of aggression in German psychiatric hospitals: Is there an increase?

**Frank Eisele[1], Erich Flammer[2], Tilman Steinert[2]\***

**1** Centers for Psychiatry Suedwuerttemberg, Ravensburg, Germany, **2** Clinic for Psychiatry and Psychotherapy I, Ulm University, Ulm, Germany

\* tilman.steinert@zfp-zentrum.de

## Abstract

### Introduction

In a meta-analysis of international studies, 17% of admitted patients in psychiatric hospitals had exhibited violent behavior toward others. Reported data from studies in Germany were considerably lower until recent years. However, studies examining only single hospitals, as well as the quality of the data itself, have raised questions as to the validity of these findings. Indeed, a debate currently exists as to whether there has, in fact, been an increase of violent incidents in German mental institutions.

### Methods

In a group of 10 hospitals serving about half the population of the Federal State of Baden-Wuerttemberg with 11 million inhabitants, the Staff Observation Aggression Scale–Revised (SOAS-R) was introduced into patients' electronic charts as part of routine documentation. Data recording was strongly supported by staff councils and unions. A completed data set is now available for the year 2019. For one hospital, data are available since 2006. Due to some doubts with respect to fully covering self-directed aggression, we restricted the analysis to aggression toward others and toward objects.

### Results

In 2019, 17,599 aggressive incidents were recorded in 64,367 admissions (1,660 staying forensic psychiatric inpatients included). 5,084 (7.90%) of the admitted cases showed aggressive behavior toward others. Variation between hospitals was low to modest (SD = 1.50). The mean SOAS-R score was 11.8 (SD between hospitals 1.20%). 23% of the incidents resulted in bodily harm. The percentage of patients showing violent behavior was highest among patients with organic disorders (ICD-10 F0) and lowest among patients with addictive or affective disorders (F1, F3, F4). Forensic psychiatry had the highest proportion of cases with aggressive behavior (20.54%), but the number of incidents per bed was lower than in general adult psychiatry and child and adolescent psychiatry (indicating a lower risk for staff). In the hospital with longer-term recordings available, an increase could be observed since 2010, with considerable variation between years.

**Data Availability Statement:** Data are available from Dryad (doi:10.5061/dryad.xpnvx0kdq). The data are anonymized and do not allow identification of patients or hospitals.

**Funding:** The authors received no specific funding for this work.

**Competing interests:** The authors have declared that no competing interests exist.

## Conclusions

This is the most robust estimate of the frequency of violent incidents in German psychiatric hospitals thus far. The incidence is about half of what has been reported internationally, probably due to sample selection bias in previous studies and a relatively high number of hospital beds in Germany. Available data suggest an increase of violent incidents over the last ten years; however, it is unclear to which extent this is due to increased reporting.

## Introduction

Workplace violence is a major problem for professionals in mental health services [1], comparable to other professionals in emergency and health services such as police, firefighters, and paramedics [2]. While violence in psychiatric institutions was previously considered as a specific psychiatric problem, caused by patients who are not responsible for their dangerous actions due to their mental illness [3], it is nowadays rather considered as a societal phenomenon that occurs in conflict situations in schools, social services, emergency rooms, and hospitals. Violent crime has been more or less continuously decreasing in the Western world for decades [4, 5]; however, violence against paramedics, figherfighters, and medical staff has been receiving increasing attention, is designated as "unacceptable" by the World Health Organization [6], and is a frequent phenomenon across countries [7]. It is unclear whether this is due to a real increase or rather to an increased propensity to report such incidents. Long-term, large-scale studies using identical instruments for measurement are lacking. Yet the management of violence is of paramount importance for psychiatric hospitals, for several reasons. First, it is the employer's duty to protect staff from forseeable risks, and violence by patients is the major workplace risk for staff on psychiatric wards. Similarly, fellow patients may feel intimidated and suffer from an environment perceived as unsafe and potentially traumatizing [8]. Second, workplace violence contributes to burnout among mental health workers and can lead to loss of qualified staff [9]. Third, violence is inevitably intertwined with the ugly face of psychiatry, the use of coercion. Violent behavior is the most frequent justification of coercive interventions under clinical and legal aspects as well [10]. All efforts to reduce coercive interventions can be successful as much as they are effective in reducing aggressive and violent behavior [11]. Staff, understandably, will be reluctant to reduce the use of coercion if the cost is an increase in violence [12, 13]. Hence, it is absolutely necessary for purposes of clinical evaluations and safety to record both the frequency of violent incidents and coercive interventions in psychiatric hospitals [14].

In a meta-analysis of studies on patients admitted to acute psychiatric units in high-income countries, Iozzino et al. [15] included 35 studies with 23,972 inpatients from 12 countries hospitalized between 2005 and 2014. Of these patients, a mean of 17% (range 3% to 44%) committed at least one act of violence. The proportion of violent patients was significantly greater in subgroups of studies rated as being of inferior study quality. Germany was represented in this meta-analysis by only one study from a single hospital, reporting a significantly lower percentage of 7.7% violent patients [16]. Generally, in contrast to the high clinical relevance and emotional involvement of victims of workplace-related violence, empirical data on violence in psychiatric hospitals in Germany is scarce thus far. To the best of our knowledge, in addition to the study by Ketelsen et al. [16], only a very limited number of relevant papers comprising at least total hospital populations have been published, and all others only in the German language. This dearth of literature causes some problems for systematic reviews [17]. A first study

published thirty years ago reported low rates of 1.9% of patients with violent acts in four hospitals in Baden-Wuerttemberg [18]. A similar result emerged from a hospital in Bavaria, covering over six years with 2.7% of patients with aggressive behavior during hospitalization [19]. From another hospital in Bavaria, a rate of 6.0% was reported for the years 1996–2001 [20]. The rate of 7.7% reported from a hospital in Northrhine Westfalia some years later by Ketelsen et al. [16] was still higher. More recently, Mueller et al. [21] reported a significant increase of violent incidents between 2008 and 2015 in a hospital in the State of Hesse, but on a rather low level (from 1.78% up to 3.32% of all admissions). In a subsequent article considering the following year, 2016, a pattern of increase could not be confirmed [22]. Whether there has been an increase in inpatient violence in German-speaking countries was first discussed by Schanda and Taylor in 2001 [23]. Referring to data from the US and the UK, they pointed out that, to date, the problem had received less attention in German-speaking countries. Since then, it is a recurring topic in debates among clinical staff, management, and staff councils. However, except for the data mentioned above from single hospitals with a high variance among them, there are no data available to answer epidemiological questions. Therefore, we set out to implement a valid and reliable system of data recording in a hospital group providing more than 3,000 hospital beds in southwest Germany. The research questions were:

1. In which percentage of admitted patients do aggressive incidents occur?

2. Are there differences between general psychiatry, geriatric psychiatry, child and adolescent psychiatry, and forensic psychiatry?

3. Are there differences with respect to psychiatric diagnoses?

4. Is there a change of the frequency of violent behaviors over time?

## Methods

### Setting

The Centers for Psychiatry (ZfP Group Baden-Wuerttemberg) comprise 9 psychiatric hospitals, each of which has several satellite centers, for a total of 38 sites. They provide 3,494 beds for inpatient treatment and 612 places in day clinics, as well as 997 places in 8 clinics for forensic psychiatry. They employ a total of 9,920 staff members; the mean total length of stay (except for forensic psychiatry) is 25.3 days. In the forensic hospitals, length of stay is about five years, varying between hospitals and across time. The 9 centers are organized in three separate companies, all of which are state-run and share common administrative structures. Our data set includes data from another hospital in which one of the centers holds shares. This hospital provides 127 beds for inpatient treatment and 85 places in day clinics and employs a total of 304 persons. The 10 hospitals serve defined catchment areas and thus are responsible for the inpatient mental health care of about half the population of the Federal State of Baden-Wuerttemberg (with 11 million inhabitants) and all the State's forensic hospital beds. As the legal conditions of hospital staffing and availability of hospital beds are more or less the same within the whole federal state, the sample can be considered as approximately representative of the state in totality.

### Measurement

The Staff Observation Aggression Scale–Revised (SOAS-R) is the most widely used instrument in Europe for the recording and measurement of aggressive inpatient behavior. It has proved to have good validity and reliability [24–26]. The SOAS was constructed to assess the characteristics and severity of aggressive and/or violent acts of psychiatric inpatients. Aggression is

defined as any verbal, nonverbal, and/or physical behavior that is either threatening or actually harmful to persons or property. There are operational criteria in the SOAS subscales that can be used to assess the severity of an event. The SOAS-R assesses five separate and consecutive aspects of aggressive incidents. The first and the last of these five aspects, respectively, describe the immediate cause and the measures taken to stop an act of aggression. The three central aspects characterizing the aggressive incident describe means, aims, and consequences. The items are rated on a 5-point Likert scale ranging from 0 to 4. The SOAS-R total score enables the classification of aggressive events as mild (score 1–5), moderate (score 6–8), or severe (score $\geq$ 9); the maximum score is 22. The SOAS-R format is limited to one page; completing the instrument takes only about two minutes. Interrater reliability was shown to be good, even without any previous training or education of the staff [19]; this finding was replicated in subsequent studies (in German, [16]). In 2004, we had translated the original version of the SOAS-R into German and re-translated it into English, assessing the result by independent experts. This version was incorporated into the electronic charts of one hospital and subsequently adopted by others.

## Implementation

The implementation of the SOAS-R into the electronic charts was accomplished in one hospital in 2006. The SOAS-R is part of the patients' medical files and can be completed for every incident by everyone who has access to the respective file (usually all team members of the respective ward and the on-duty physician). Consecutively, an increasing number of hospitals introduced the SOAS-R beginning from 2010; in 2016, the hospital managers decided to introduce it on an obligatory basis in all hospitals for all wards. For this purpose, a common codebook with clear definitions and descriptions was developed and distributed among the hospitals. Complete datasets are available for 7 hospitals since 2017 and for all 10 hospitals since 2018. In the beginning, staff members on the wards were somewhat unwilling (as often occurs with the introduction of additional forms and bureaucracy)—despite the instrument's ease and rapidity of completion. This, in all likelihood, had led to underreporting of aggressive incidents in the initial years after 2006. However, this changed when staff councils realized that statistics generated from the SOAS files documented the difficulties and dangerousness of their work. Since then, representatives of the staff council have been regularly appealing to staff working on the wards and encouraging them to complete the SOAS-R when aggressive incidents occur. Based on the SOAS-R statistics, staff councils consider annually, in concert with the responsible managerial boards, which preventive measures should be taken to protect staff from violence, taking into account the German work safety law (Arbeitsschutzgesetz). As the SOAS-R is viewed and has been introduced as an instrument to record patients' aggressive behavior towards others, particularly staff, we have some doubts regarding whether self-directed aggressive behavior, which was not the objective of this study, is recorded completely.

## Data structure

The 10 hospitals each provided three datasets. Dataset 1 contains the aggressive incidents (one dataset for each incident), together with all the items of the SOAS-R questionnaire, the hospital name, pseudonymized case numbers, gender, main diagnosis, and the legal basis for the hospital stay. The other two datasets contain only aggregated data on the number of treated cases and treatment duration. The data are thus structured in such a way that the identification of specific persons is not possible, i.e. the data are anonymized.

The used data refer to cases, not to patients. Cases are defined as discharges in a reporting year, irrespective whether the admission occurred in the previous or in the current reporting

year. Because of readmissions, the number of patients is lower than the number of cases. Due to sometimes very long durations of hospital stays in forensic psychiatry, a different case definition was used there. In forensic psychiatry, all patients were included who had stayed there at least one day in the reporting year, regardless of the year in which they had been admitted or discharged.

## Analysis

We present the data from the complete year 2019. We excluded cases with only self-directed aggression (3.4% of reported cases) due to suspected underreporting. As outcomes, we determined the proportion of cases in which at least one SOAS-R form was recorded, the mean number of incidents per case with aggressive behavior, and the mean SOAS-R score. Results were divided 1) per speciality department (general psychiatry, geriatric psychiatry, child and adolescent psychiatry, forensic psychiatry), and 2) per main diagnosis according to ICD-10. To analyse differences across hospitals, we determined mean and standard deviations, as well as the median values and ranges.

## Ethics

The Ethics Committee of Ulm University waived the requirement for ethics approval as approval is not required for retrospective studies analyzing anonymized data, in accordance with national legislation and institutional requirements.

## Results

### 1. Cross-sectional analysis

In total, in 2019, there were 62,707 cases discharged from the psychiatric hospitals and 1.660 inpatients treated in forensic psychiatry. In 5,084 (7.90%, SD between hospitals = 1.50%) of these 64,367 cases, aggressive incidents were recorded. The total number of aggressive incidents was 17,599, with a mean SOAS-R score of 11.8 (SD = 1.20). In those cases with aggressive behavior, on average, 3.46 (SD = 0.91) incidents occurred. 23% (SD between hospitals = 9.45%) of the aggressive incidents resulted in physical harm. Table 1 shows the frequency of aggressive incidents in the 10 hospitals, Table 2 shows the frequency of aggressive incidents in the different types of specialty departments.

The aggressive incidents were directed against staff in 74.8% of total cases reported, toward objects (including arson) in 15.2%, and against fellow patients in 22.8%; multiple answers were possible. Table 3 shows the distribution of aggressive incidents according to diagnoses

**Table 1. Frequency of aggressive incidents in psychiatric hospitals.**

| Hospitals | Total number of cases N | Number of aggressive incidents | Proportion of cases with aggressive incidents | Mean SOAS-R score |
|-----------|-------------------------|--------------------------------|-----------------------------------------------|-------------------|
| Hosp 1 | 7,981 | 984 | 4.21% | 12.3 |
| Hosp 2 | 8,457 | 1,661 | 8.19% | 9.9 |
| Hosp 3 | 2,773 | 776 | 6.13% | 10.6 |
| Hosp 4 | 4,966 | 1,113 | 7.23% | 11.2 |
| Hosp 5 | 3,154 | 1,418 | 8.81% | 11.8 |
| Hosp 6 | 8,118 | 1,657 | 8.47% | 11.5 |
| Hosp 7 | 6,556 | 1,870 | 9.37% | 10.9 |
| Hosp 8 | 10,495 | 4,505 | 9.10% | 11.7 |
| Hosp 9 | 8,423 | 2,508 | 8.29% | 13.6 |
| Hosp 10 | 3,444 | 1,107 | 8.54% | 14.0 |

**Table 2. Frequency of aggressive incidents in psychiatric inpatient facilities.**

| | Total number of cases N | Proportion of cases with aggressive incidents | Standard deviation between 10 hospitals | Median (range) | Mean SOAS-R score (SD) |
|---|---|---|---|---|---|
| Adult psychiatry | 57,350 | 7.95% | 0.016 | 8.59% (4.12%–9.42%) | 11.91 (1,22) |
| Child and adolescent psychiatry | 2,223 | 8.10% | 0.016 | 8.80% (6.21%–10.14%) | 11.55 (0,69) |
| Forensic Psychiatry | 1,660 | 20.54% | 0.087* | 23.59% (31.60% – 5.29%) | 10.79 (1,11) |
| Psychosomatics | 3,134 | 0.19% | 0.003 | 0.00 (0.98%–0.00%) | 4.67 (3,25) |

* Forensic psychiatry is available in only 8 of 10 hospitals.

(ICD-10 main clinical diagnosis) in the clinical departments except for forensic psychiatry, and Table 4 shows the distribution of aggressive incidents according to ICD-10 diagnoses in forensic psychiatry.

## Longitudinal analysis

In the hospital where the SOAS-R had been implemented already in 2006 (Hosp 7), we observed the numbers of incidents resulting in physical harm (threshold defined as pain > 10 min.), as indicated in Table 5.

## Discussion

The present survey, incorporating over 60,000 admissions per year in a representative population of psychiatric hospitals and, in addition, longitudinal data, is by far the largest examination of inpatient violence in psychiatric facilities available so far. Indeed, it exceeds by about threefold the total number of cases from 10 countries in the only meta-analysis that had previously been published [15]. Regarding our first research question, due to the high number of included cases, clearly defined method, and thorough implementation over several years, the estimate that about 8% of admissions show aggressive behavior against others seems rather robust. This is underpinned by the rather low variance across the 10 hospitals, each with an

**Table 3. Frequency of aggressive incidents according to diagnoses (adult psychiatry, child and adolescent psychiatry and psychosomatics, autoaggression excluded).**

| | Total number of cases N | Proportion of cases with aggressive incidents (SD) | Mean SOAS-R score (SD) | Mean number of incidents per case with recorded aggression |
|---|---|---|---|---|
| Organic disorders (F0/G30) | 5,379 | 26.25% (5.82%) | 12.22 (1.29) | 4.53 |
| Addictive disorders (F1) | 20,285 | 3.12% (1.48%) | 10.06 (1.49) | 1.63 |
| Schizophrenic disorders (F2) | 11,215 | 14.69% (3.49%) | 11.79 (1.08) | 3.37 |
| Affective disorders (F3) | 16,737 | 2.81% (1.07%) | 11.04 (1.06) | 2.68 |
| Neurotic, stress-related and somatoform disorders (F4) | 4,936 | 2.51% (0.76%) | 12.07 (2.20)Ag | 2.16 |
| Behavioral syndromes associated with physiological disturbances and physical factors (F5) | 217 | 0.92% (5.20%) | 14.00 (1.33) | 2.00 |
| Personality disorders (F6) | 2,445 | 8.47% (3.14%) | 11.54 (2.11) | 2.82 |
| Mental retardation (F7) | 348 | 21.55% (14.48%) | 15.29 (2.27) | 2.89 |
| Disorders of psychological development (F8) | 117 | 21.37% (20.35%) | 15.33 (4.43) | 14.08 |
| Behavioral and emotional disorders with onset usually occurring in childhood and adolescence (F9) | 693 | 12.12% (7.03%) | 10.92 (1.40) | 2.45 |
| Other | 335 | 19.10% (23.61%) | 11.69 (3.86) | 3.36 |

**Table 4. Frequency of aggressive incidents according to diagnoses (only forensic psychiatry, autoaggression excluded).**

|  | Total number of cases N | Proportion of cases with aggressive incidents (SD) | Mean SOAS -R score (SD) | Mean number of incidents per case with recorded aggression |
|---|---|---|---|---|
| Organic disorders (F0/G30) | 32 | 46.88% (20.87%) | 10.91 (4.39) | 7.20 |
| Addictive disorders (F1) | 728 | 10.03% (10.00%) | 9.24 (4.09) | 1.96 |
| Schizophrenic disorders (F2) | 658 | 26.60% (14.19%) | 11.00 (3.87) | 4.19 |
| Affective disorders (F3) | 25 | 16.00% (18.84%) | 10.13 (5.32) | 9.75 |
| Neurotic, stress-related and somatoform disorders (F4) | 1 |  |  |  |
| Behavioral syndromes associated with physiological disturbances and physical factors(F5) | 1 |  |  |  |
| Personality disorders (F6) | 119 | 27.73% (13.82%) | 10.46 (4.31) | 4.52 |
| Mental retardation (F7) | 68 | 51.47% (18.80%) | 10.54 (2.23) | 5.37 |
| Disorders of psychological development (F8) | 10 | 40.00% (25.00%) | 12.14 (5.07) | 34.00 |
| Behavioral and emotional disorders with onset usually occurring in childhood and adolescence(F9) | 7 | 14.29% (40.00%) | 10.00 (4.00) | 13.00 |
| Other | 11 | 9.09% (12.42%) | 11.00 (4.10) | 3.00 |

own catchment area—though they clearly have some relevant differences, for example, with regard to the percentage of involuntary admissions [27]. The only hospital with a significantly lower percentage of recorded cases (Hosp 1, Table 1) had implemented data collection only in 2018, with a significant increase in 2019. Since figures had developed similarly in other hospitals after the initial implementation, we strongly believe that this outlier is not caused by hospital or patient characteristics but rather by underreporting. Noticeably, the highest rate was observed in Hosp 7, where the SOAS-R had been introduced already in 2006. Generally, underreporting seems conceivable with respect to violence against fellow patients that might escape the attention of staff. At least, our findings of a percentage of 22.8% incidents toward

**Table 5. Longitudinal data on the frequency of aggressive incidents with physical harm in one hospital 2006–2019.**

|  | 2006 | 2007 | 2008 | 2009 | 2010 | 2011 | 2012 | 2013 | 2014 | 2015 | 2016 | 2017 | 2018 | 2019 |
|---|---|---|---|---|---|---|---|---|---|---|---|---|---|---|
| **General psychiatry** |  |  |  |  |  |  |  |  |  |  |  |  |  |  |
| Aggressive incidents | 101 | 107 | 115 | 97 | 139 | 141 | 150 | 178 | 241 | 224 | 164 | 164 | 251 | 302 |
| Number of cases | 3,066 | 3,244 | 3,196 | 3,367 | 3,508 | 3,875 | 4,077 | 4,223 | 4,608 | 4,444 | 4,589 | 4,544 | 4,320 | 4,518 |
| Aggressive incidents per treated cases | 0.03 | 0.03 | 0.04 | 0.03 | 0.04 | 0.04 | 0.04 | 0.04 | 0.05 | 0.05 | 0.04 | 0.04 | 0.06 | 0.07 |
| Proportion of cases with aggressive incidents | 1.73% | 1.91% | 1.41% | 1.34% | 1.68% | 2.19% | 2.08% | 1.59% | 2.24% | 1.71% | 1.72% | 1.96% | 1.83% | 2.66% |
| **Child and adolescent psychiatry** |  |  |  |  |  |  |  |  |  |  |  |  |  |  |
| Aggressive incidents | 6 | 22 | 10 | 9 | 3 | 41 | 39 | 81 | 209 | 105 | 153 | 124 | 126 | 206 |
| Number of cases | 496 | 516 | 590 | 623 | 654 | 691 | 685 | 734 | 818 | 815 | 896 | 743 | 789 | 759 |
| Aggressive incidents per treated cases | 0.01 | 0.04 | 0.02 | 0.01 | 0.00 | 0.06 | 0.06 | 0.11 | 0.26 | 0.13 | 0.17 | 0.17 | 0.16 | 0.27 |
| Proportion of cases with aggressive incidents | 0.40% | 1.94% | 1.69% | 1.28% | 0.46% | 2.75% | 4.23% | 5.59% | 5.38% | 4.79% | 6.81% | 6.33% | 6.59% | 8.30% |
| **Geriatric psychiatry** |  |  |  |  |  |  |  |  |  |  |  |  |  |  |
| Aggressive incidents | 10 | 15 | 22 | 33 | 55 | 55 | 45 | 48 | 50 | 63 | 45 | 70 | 50 | 52 |
| Number of cases | 602 | 572 | 598 | 580 | 558 | 571 | 601 | 578 | 644 | 698 | 684 | 692 | 756 | 691 |
| Aggressive incidents per treated cases | 0.02 | 0.03 | 0.04 | 0.06 | 0.10 | 0.10 | 0.07 | 0.08 | 0.08 | 0.09 | 0.07 | 0.10 | 0.07 | 0.08 |
| Proportion of cases with aggressive incidents | 1.66% | 2.45% | 2.84% | 4.31% | 5.73% | 6.30% | 4.83% | 4.15% | 5.12% | 4.30% | 4.09% | 5.64% | 4.89% | 5.50% |
| **Forensic psychiatry** |  |  |  |  |  |  |  |  |  |  |  |  |  |  |
| Aggressive incidents | 1 | 2 | 4 | 9 | 4 | 3 | 0 | 4 | 1 | 5 | 8 | 4 | 24 | 29 |
| Number of cases | 145 | 140 | 160 | 161 | 167 | 160 | 175 | 174 | 169 | 167 | 155 | 168 | 175 | 192 |
| Aggressive incidents per treated cases | 0.01 | 0.01 | 0.03 | 0.06 | 0.02 | 0.02 | 0.00 | 0.02 | 0.01 | 0.03 | 0.05 | 0.02 | 0.14 | 0.15 |
| Proportion of cases with aggressive incidents | 0.69% | 1.43% | 1.88% | 4.97% | 1.80% | 1.88% | 0.00% | 2.30% | 0.59% | 2.99% | 5.16% | 1.79% | 9.14% | 8.33% |

fellow patients are well in line with an US hospital survey [28] that found a proportion of 20.7% of incidents directed toward other patients. Our estimates of the incidence of violence are well in line with the previous work of Ketelsen et al. in Germany [16] who, 15 years ago, found an incidence of 7.7% in a hospital located in a different federal state. They are considerably higher than the reported figures from other studies in Germany [18–21]. The most probable reason is underreporting in these both older and more recent studies due to incomplete implementation of the recording system.

Regarding our second research question, the comparison between different types of facilities showed that the percentage of violent patients was highest in forensic psychiatry, followed by child and adolescent psychiatry and adult psychiatry, and lowest, as expected, in psychosomatics. However, though forensic psychiatry expectedly not only had the highest percentage of violent patients and the highest rate of observed aggressive incidents per patient, the incidence of violent assaults per bed was lower on forensic psychiatric wards than on general psychiatric wards. This is due to the fact that a hospital bed is occupied by not much more than one patient per year, but by about 15 patients in general psychiatry. A presentation of results related to hospital beds instead of cases, representing the staff perspective, would have yielded different results (showing that working in forensic psychiatry is comparably safe). Also, the percentage of cases with a consequence of physical harm was lowest in forensic psychiatry.

Regarding our third research question of differences between diagnoses, by far the highest percentage of cases with aggressive behavior was observed among cases with ICD-10 diagnoses of F0 (organic disorders) and F7 (low intelligence). This is not surprising given that, in most cases, these disorders represent persistent states that cannot be treated directly and that are not a reason for hospital admission. Hospital admission typically occurs as a consequence of severe disorders of behavior, first of all, of violent nature. As well, in US hospitals, violent incidents in geriatric units are reported to be five times higher than in general psychiatric units [28]. Among affective disorders (ICD-10 F3), it was not possible to distinguish between depressive states with a probably little incidence of violent behavior and manic states with a probability of a substantially higher incidence of assaults. This explains why the percentages of violent patients are considerably lower than among patients with schizophrenic disorder (ICD-10 F2).

With respect to the last research question (which is most intensely under discussion), our results from one hospital, across 14 years, yielded some evidence of an increase of violence in psychiatric hospitals, except for geriatric psychiatry. There has been an increase between 2006 and 2010; however, this development was always observed in subsequent hospitals after implementation of the new reporting system for some years. A very plausible explanation for this increase is an incomplete recording in the beginning. This applied also for Hospital 1 (presenting with the lowest proportion of cases with violent incidents, where the SOAS-R recording had been introduced latest). However, even after 2010, there was a significant increase, from 0.04 incidents/case in 2010 to 0.07 in 2019 in general psychiatry, considerably more in child and adolescent psychiatry and forensic psychiatry, but not in geriatric psychiatry, where the rates held rather consistently at a high level. On the other hand, the rate in general psychiatry in 2016 was the same as in 2010; each future year could indicate a change of any trend. Hence, we cannot determine with certainty whether the registered increase of incident reporting reflects a real increase of violence or whether it is due to improved reporting. Reasons for improved reporting might be not only of administrative character. Even if we had introduced a validated instrument by using the SOAS-R, the subjective perception of violence and, associated with it, the threshold to fill a form, might have changed within the observed years due to increased public awareness. While in former years, staff in psychiatric facilities frequently viewed at least milder forms of violence by inpatients as "part of the job," nowadays, each kind of violence is considered unacceptable [6]; rather, an attitude of "zero tolerance" prevails.

There remains the question of whether and why the percentage of violent inpatients should be lower by more than one half of that reported in studies from other high-income countries [15]. Looking into the method of that systematic review, it turns out that all studies referring to "acute psychiatric wards" had been included. This definition causes a major sample selection bias. In some countries, particularly those with a relatively small number of hospital beds per population (for example, Italy and UK), there is no difference between "acute psychiatric wards" and "psychiatric wards," since only "acute" patients are admitted to hospitals, with a high proportion of them being admitted on an involuntary basis. In other countries such as Germany, Switzerland, or the Netherlands, a considerably higher number of hospital beds per population allows for a internal differentiation of hospitals with "acute" wards, rehabilitation wards, specialized wards for depressive disorders, psychotherapy, and so on. The degree of differentiation is mostly dependent on the amount of available beds (a figure that differs considerably across hospitals). Who is admitted where follows rules of the respective hospital organization and is not comparable among hospitals. Therefore, studies including only "acute" wards, not further defined, are not useful for any epidemiological calculations [29–32]. Instead, epidemiological calculations of the incidence of inpatient violence should necessarily be based only on data on total hospital admissions, so as to avoid sample selection bias (such as in our study and the previous German studies mentioned above). Hence, the frequently-cited meta-analysis by Iozzino et al. [15] has considerably overestimated the problem due to sample selection bias.

Another reason why the proportion of violent inpatients might indeed be lower in Germany than in some other countries such as the UK or Italy is, as mentioned, the higher number of available hospitals beds. For example, the number of psychiatric beds in general hospitals and in mental hospitals per 100,000 population is indicated as 136.3 in Germany, 89.4 in Switzerland, 29.8 in the US, 29.3 in France, 23.9 in the UK, and 9.0 in Italy [33]. However, addiction psychiatry and geriatric psychiatry are not counted among psychiatric beds in some of these countries. A considerable number of admitted patients in Germany (see Table 3) suffer from depressive and adjustment disorders, are treated on a voluntary basis, and only rarely exhibit violence. In countries with a low number of available hospital beds, a considerable proportion of these patients probably would have not been admitted as inpatients. With respect to countries with low and middle income where inpatient psychiatric treatment frequently takes place in general hospitals and not in separated psychiatric facilities, we cannot draw any conclusions from the data.

Our study has two major limitations. First, notwithstanding the considerable efforts in terms of implementation and robust data over years and across numerous hospitals, underreporting can never be excluded in routine data. As the SOAS-R reporting system is widely perceived as a documentation system for violence against hospital employees, it can be assumed that this applies more to violence between patients [29] than to violence toward staff. The second limitation is inherent in the SOAS-R and all other instruments that classify aggressive incidents by a score. It is not ascertained that these scores provide valid measures of severity in all cases. For example, a threat with a weapon by a young paranoid patient, but eventually without any physical harm, can score lower than a patient with dementia hitting a nurse during assistance in bathing or grooming. Therefore, we do not believe that our data prove that geriatric psychiatry is the most dangerous working place in psychiatric facilities, even if suggesting so on face.

## Conclusions

This is the most robust estimate of the frequency of violent incidents in German psychiatric hospitals conducted thus far. The incidence is about half of what has been reported

internationally, probably due to sample selection bias in previous studies and a relatively high number of hospital beds in Germany. Available data suggest an increase of violent incidents over the last 10 years; however, it is unclear to which extent this is due to increased reporting.

## Author Contributions

**Conceptualization:** Tilman Steinert.

**Data curation:** Frank Eisele.

**Formal analysis:** Erich Flammer.

**Methodology:** Frank Eisele.

**Resources:** Frank Eisele.

**Validation:** Frank Eisele, Erich Flammer.

**Writing – original draft:** Tilman Steinert.

**Writing – review & editing:** Frank Eisele, Erich Flammer, Tilman Steinert.

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
