## [Decision Letter · Decision Letter 0]

2 Dec 2020

PONE-D-20-32386

Aggressive Incidents in German Psychiatric Hospitals: Is there an Increase?

PLOS ONE

Dear Dr. Steinert,

Thank you for submitting your manuscript to PLOS ONE. After careful consideration, we feel that it has merit but does not fully meet PLOS ONE’s publication criteria as it currently stands. Therefore, we invite you to submit a revised version of the manuscript that addresses the points raised during the review process.

We look forward to receiving your revised manuscript.

Kind regards,

Stephan Doering, M.D.

Academic Editor

PLOS ONE

Reviewers' comments:

Reviewer's Responses to Questions

**Comments to the Author**

1. Is the manuscript technically sound, and do the data support the conclusions?

Reviewer #1: Yes

Reviewer #2: Yes

2. Has the statistical analysis been performed appropriately and rigorously? 

Reviewer #1: Yes

Reviewer #2: Yes

3. Have the authors made all data underlying the findings in their manuscript fully available?

Reviewer #1: Yes

Reviewer #2: No

4. Is the manuscript presented in an intelligible fashion and written in standard English?

Reviewer #1: Yes

Reviewer #2: No

5. Review Comments to the Author

Reviewer #1: The manuscript clearly describes the different sections of methodology, results and conclusions, responding to the proposed objectives.

It is written clearly and simply, easy to read.

There are 3 things I would like to review:

In the first place it would be good to comment in the text that there is evidence of a generalized increase in aggressions in society in general and especially in the workplace, but especially in the health sector. In addition, there are studies that show that this violence is growing in all health areas, not only in psychiatry, there are even authors who assure that there is a higher rate of aggression in other units, such as in primary care. I think it would be good to mention this and then talk specifically about assaults in psychiatry.

Secondly, in the discussion, I believe that the point where the possible bias is discussed when comparing the results with those of other countries should be developed a little more, since psychiatry is not so developed in all countries nor are there hospitals so specific to treat these patients, these patients are sometimes in general hospital, this can make it difficult to compare with data from other countries.

And thirdly, I believe that it should also be considered that despite passing a scale to assess the aggressions received, it should be taken into account that it is a very subjective issue, since not all people perceive violence in the same way, there is Studies that show that there are health workers who take aggressions as an inherent part of their work and even more so if they are personnel who work with psychiatric patients.

Reviewer #2: The current study is an interesting multi-center study aimed to investigate whether there is an increase of violent incidents in German Psychiatric Hospitals. This is a very relevant topic, both for staff as well as for patients. In general, the text will benefit from some relatively minor language adjustments. Furthermore, the authors should emphasize on which form of aggression they included and how this form relates to previous literature.

Abstract

1. Was the SOAS-R introduced by the researchers, or by the board of each hospital as a part of care?

2. What is the rationale for the focus on aggressive behaviour towards others? The SOAS-R measures verbal aggression, physical aggression (towards others and towards objects), and aggression against oneself. You should emphasize on this in the manuscript.

Introduction

1. The authors only mention the negative impact on health care workers. However, there is also a burden for patients and their environment, making aggressive behaviour an even more relevant topic to study.

2. Of what kind are the hospitals mentioned in the introduction? Acute, long-term, forensic?

3. The introduction states “We set out to implement a valid and reliable system of data recording in a hospital group”. Although this seems a prospective study, it is mentioned later that the current study is retrospective. This is not clear from the introduction.

4. How does the second research question relate to previous studies?

Methods

1. There are several typos in the manuscript, especially in the methods (goog = good, p.6; independet = independent, p.7; assess = assesses, p.7; behavioud = behaviour, p.9; i = in, p.9).

2. What was the length of stay for forensic psychiatry?

3. How can the maximum score be 21, considering the SOAS-R has 5 aspects with a range from 0 to 4? As described in the article of Nijman et al (1999), the SOAS-R has a range from 0 to 22 and does not use a Likert scale.

4. “However, this changed when staff councils realized that statistics generated from the SOAS files documented the difficulties and dangerousness of their work.” How do you know the staff became aware? Focus groups? Otherwise, you could use a reference to state that implementing something new takes time.

5. Why exclude cases with only self-directed aggression? And how does this relate to previous estimates? Did you also exclude self-directed aggression?

Results

1. Perhaps you could show your main findings in a bar chart?

2. Did you exclude verbal aggression against others as well?

3. Why did you exclude forensic psychiatry from Table 3?

Discussion

1. I would suggest making a conclusion paragraph to end the discussion.

6. PLOS authors have the option to publish the peer review history of their article (what does this mean?). If published, this will include your full peer review and any attached files.

Reviewer #1: **Yes: **MI Serrano Vicente

Reviewer #2: No

---

## [Author Response · Author response to Decision Letter 0]

20 Dec 2020

against our previous concerns, we decided to make data publicly accessible and created a data repository as provided with the respective URL. We understood that this will be the gold standard for publications also in the future.

---

## [Editor Report · Decision Letter 1]

22 Dec 2020

Incidents of Aggression in German Psychiatric Hospitals: Is there an Increase?

PONE-D-20-32386R1

Dear Dr. Steinert,

We’re pleased to inform you that your manuscript has been judged scientifically suitable for publication and will be formally accepted for publication once it meets all outstanding technical requirements.

Kind regards,

Stephan Doering, M.D.

Academic Editor

PLOS ONE

---

## [Editor Report · Acceptance letter]

26 Dec 2020

PONE-D-20-32386R1 

Incidents of aggression in German psychiatric hospitals: Is there an increase? 

Dear Dr. Steinert:

I'm pleased to inform you that your manuscript has been deemed suitable for publication in PLOS ONE. Congratulations! Your manuscript is now with our production department. 

Kind regards, 

on behalf of

Professor Stephan Doering 

Academic Editor

PLOS ONE